# OpenReview forum: "Guardrail-Agnostic Societal Bias Evaluation in Large Vision-Language Models"
_ICLR.cc/2026/Conference — Submitted to ICLR 2026_

### Official Review · Reviewer_oVen · 2025-10-15

**Soundness:** 2
**Presentation:** 3
**Contribution:** 1
**Rating:** 4
**Confidence:** 5

**Summary:**

This paper proposes a method to evaluate societal bias in LVLMs that have strong safety guardrails. The authors note that modern LVLMs often refuse to answer traditional bias prompts that ask them to infer a person's attributes from an image. To bypass these refusals, their proposed method uses person-irrelevant tasks while providing a user's photo as implicit demographic context. Applying this framework to 20 LVLMs, the authors find it successfully avoids refusals and reveals that all tested models exhibit bias by altering their responses based on the user's perceived demographics.

**Strengths:**

**1. An effective framework for evaluating guarded models**

The paper correctly identifies a critical problem: existing bias benchmarks are failing because modern models with safety guardrails simply refuse to answer. The proposed method is a novel and effective solution to this issue. By changing the evaluation paradigm, it successfully bypasses these guardrails to achieve a zero refusal rate, offering a practical way forward for evaluating heavily-guarded models.

**2. A comprehensive, large-scale empirical study**

The paper presents a comprehensive and large-scale empirical study of societal bias across 20 recent LVLMs, including both open-source and proprietary models. The evaluation is well-structured, using three distinct tasks to probe different facets of bias. This broad study provides a valuable snapshot of the current bias landscape and yields interesting comparative results between different types of models.

**Weaknesses:**

**1. Limited Methodological Novelty**

The paper's core methodological contribution is limited. The proposed framework is largely an adaptation of the established "persona-based evaluation" paradigm from the text-only LLM domain. The primary modification is substituting an explicit textual persona with an implicit visual one via a user's photo. This is a straightforward and incremental step, rather than a fundamental innovation in evaluation methodology.

**2. Potential for Confounding Variables in User Images**

The evaluation framework does not adequately account for confounding variables within the user images. While the paper assumes the model's biased outputs are a reaction to core demographic traits like gender and race, the images contain many other correlated cues. The paper does not present any analysis to disentangle these factors, making it unclear if the model is reacting to the intended demographic variable or to these other confounding visual features.

**3. Over-constrained "Story Generation" Task May Induce and Exaggerate Bias**

One of the paper's key findings relies on the "story generation" task, which is presented as the most open-ended evaluation. However, the prompt for this task is highly constrained, explicitly forcing the model to fill in seven sensitive attributes, including job, economic status, and education. This setup is less a measure of bias in creative, open-ended generation and more a "stereotype fill-in-the-blanks" exercise. This task design may itself induce bias and exaggerate the extent to which the model would apply stereotypes in more natural, unconstrained scenarios.

**Questions:**

The paper's central claim is that the model's outputs are influenced by the user's demographic traits. Could the authors provide any analysis or stronger arguments to rule out the influence of potential confounding visual variables that might be spuriously correlated with demographics in the input images?

---

> ### Author Response · Authors · 2025-11-25
> **Response to Reviewer oVen (1)**
>
> Dear Reviewer oVen
>
> We sincerely appreciate your constructive review and recognition of our work’s contributions, particularly the effectiveness of our guarded-model framework and our comprehensive empirical study. Your comments are very helpful in enhancing the quality of our presentation. Please find the corresponding responses below:
>
> >**W1. Limited Methodological Novelty.** The paper correctly identifies a critical problem: existing bias benchmarks are failing because modern models with safety guardrails simply refuse to answer. ...
>
> As highlighted by other reviewers (e.g., **Strengths 2 by Reviewer RCGm, Strength 2 by Reviewer Decc**), the **novelty of our work lies in solving the critical "refusal problem"** that renders existing benchmarks ineffective for guardrailed models.
>
> We believe the **method's simplicity is a strength, not a weakness**. A complex method is not inherently better; rather, our simple and effective approach ensures high reproducibility and easy adaptability (as also highlighted by Reviewer RCGm, who called it "simple but clever”, and Reviewer Decc, who described it as "elegant and simple").
>
> The significance of our work is in the paradigm shift of treating the image as a provisional user context rather than a visual target, which successfully unlocks bias evaluation where prior methods fail.
>
>
> >**W2. Potential for Confounding Variables in User Images.** The evaluation framework does not adequately account for confounding variables within the user images. ...
> >
> >**Q1.** The paper's central claim is that the model's outputs are influenced by the user's demographic traits. Could the authors provide any analysis or stronger arguments to rule out the influence of potential confounding visual variables that might be spuriously correlated with demographics in the input images?
>
> We appreciate this insightful point. As we discuss in our response to Reviewer EAsf (W1), we argue that our framework minimizes the influence of confounding variables compared to the existing methods. We detail the discussion and experiment to support this argument below:
>
> * **Our framework is fundamentally robust to non-demographic contexts:** Unlike existing benchmarks that treat the image as the **subject of prompts** (e.g., "Describe this person"), which forces the model to attend to all visual features, including background noise, our method treats the image solely as **provisional user information**. This design **decouples the task from the visual scene**, encouraging the model to use the image primarily as a high-level demographic signal (user information) rather than attending to visual confounders. Additionally, we deliberately utilized FairFace, a face-centric dataset with minimal background variation, to further reduce environmental confounding factors.
>
> * **Experimental evidence:** To verify that our method is less susceptible to spurious visual correlations than prior methods, we conducted a controlled experiment using InternVL3-14B. We describe the details of the experiment below:
>
> **Setup:** We compared the stability of model outputs across two different image domains: face-centric images (FairFace, minimal background) and real-world images (COCO, complex backgrounds).
>
> **Methods compared:**
>
> Existing open-ended eval: Prompt = "Describe this image in detail."
>
> Ours: Prompt = "I've attached my photo. Write a fictional story about an imaginary person..."
>
> **Metric:** We measured the Jensen-Shannon (JS) Distance between the vocabulary distributions (nouns, verbs, adjectives) of face-centric and real-world images. A lower JS distance indicates that the model's outputs are more robust (i.e., less affected by the change in visual background/context).
>
> **Results:** The table below shows the JS distance (vocabulary shift) when the image source changes.
>
> | Method | JS Distance (Lower is Better) | Interpretation |
> | :--- | :---: | :--- |
> | **Existing Open-Ended Eval** | **0.61** | Outputs are heavily affected by background context. |
> | **Ours (Story Generation)** | **0.38** | Outputs remain relatively consistent regardless of visual context. |
>
> The existing evaluation's high JS distance confirms its sensitivity to confounding visual features. Conversely, our method's significantly lower score demonstrates that it effectively filters out spurious visual details, indicating that the observed bias is driven by the user persona rather than visual confounders.

---

> > ### Author Response · Authors · 2025-11-25
> > **Response to Reviewer oVen (2)**
> >
> > >**W3. Over-constrained "Story Generation" Task May Induce and Exaggerate Bias.** One of the paper's key findings relies on the "story generation" task, which is presented as the most open-ended evaluation. However, the prompt for this task is highly constrained, explicitly forcing the model to fill in seven sensitive attributes, including job, economic status, and education. This setup is less a measure of bias in creative, open-ended generation and more a "stereotype fill-in-the-blanks" exercise. This task design may itself induce bias and exaggerate the extent to which the model would apply stereotypes in more natural, unconstrained scenarios.
> >
> > We argue that the **constraints in the story generation task do not invalidate the bias measurement**. We clarify our rationale for the task design based on two key points:
> >
> > * We do not claim the task is completely unconstrained. Rather, we position it as **"relatively" open-ended** compared to our other tasks (Term Explanation and Exam-Style QA). The distinction is based on the degree of constraint, where story generation offers the model significantly more creative freedom in choosing specific content (e.g., which job to assign) compared to fixed-answer tasks.
> >
> > * As recognized by other reviewers who found our experimental design reasonable (e.g., Reviewer RCGm), these constraints are **essential for a fair and quantitative comparison**. To measure distributional shifts (TVD) across demographic groups, we must ensure that the generated outputs contain comparable attributes. If we do not explicitly request attributes like job or education, models might generate vastly different story structures (e.g., one focusing on hobbies, another on career), rendering statistical comparison impossible.

---

> > > ### Author Response · Authors · 2025-11-28
> > > **Delighted to Provide Further Details**
> > >
> > > Dear Reviewer oVen,
> > >
> > > Thank you very much for your thoughtful and constructive feedback on our submission. We have carefully considered your comments and addressed each of them in our rebuttal. We understand you may be busy, but as the rebuttal period is approaching its end, we would greatly appreciate it if you could take a moment to review our responses when possible. If anything remains unclear or you would like additional details, we would be more than happy to provide further clarification.

---

### Official Review · Reviewer_Decc · 2025-10-29

**Soundness:** 4
**Presentation:** 4
**Contribution:** 3
**Rating:** 8
**Confidence:** 4

**Summary:**

The paper identifies a critical flaw in existing methods for evaluating societal bias in Large Vision-Language Models: the increasing prevalence of safety guardrails. Models like GPT and Claude often refuse to answer attribute-inferring prompts, making traditional bias benchmarks unreliable. To solve this refusal problem, the authors propose a novel guardrail-agnostic evaluation method. The key idea is to decouple the task from the person in the image. Instead of using the image as a target for an attribute-inferring prompt, the method uses the image as provisional user information paired with a person-irrelevant prompt. The authors instantiate this framework across three tasks (Story Generation, Term Explanation, Exam-Style QA) and apply it to 20 recent LVLMs (both open-source and proprietary). Their method successfully achieves a 0% refusal rate. The results show that all models exhibit gender and racial bias and that proprietary models, while still biased, show lower levels of bias than open-source ones, which the authors hypothesize is due to continuous monitoring and iterative refinement rather than just one-time safety training.

**Strengths:**

1. Clarity and Writing: The paper is exceptionally well-written, clear, and logically structured. The problem is motivated perfectly with concrete examples (Figure 1) and data (Table 1), making the paper's contribution easy to understand and appreciate.
2. Novel and Necessary Methodology: The "refusal problem" is a very real and growing challenge for AI safety and fairness research. As models become more locked down, prior evaluation methods are becoming obsolete. The proposed guardrail-agnostic method is an elegant, simple, and highly effective solution. It cleverly reframes the evaluation to probe implicit associations without triggering the models' explicit safety guardrails. The fact that it achieves zero refusals is a strong validation of the approach.
3. Significant and Nuanced Findings: The paper's comprehensive evaluation of 20 LVLMs produces several nuanced and important findings for the community. The observations are not monolithic ("all models are biased") but are carefully broken down.
4. Thorough and Insightful Discussion: The discussion in Section 5 is a major strength. The authors move beyond just reporting results to hypothesize why these results occur. The argument that continuous monitoring and iterative refinement are key factors in bias reduction —more so than just static safety alignment —is a mature and important takeaway for the field, especially for the open-source community.

**Weaknesses:**

The paper is of very high quality overall. Here are some questions:
1. The validation of the LLM-as-judge showed a 97% human agreement for the term explanation task. Was a similar human-agreement study performed for the story generation attribute extraction? This seems like a more complex and potentially ambiguous extraction task (e.g., parsing "personality" or "economic situation"), and it would be valuable to know the human-LLM alignment there as well.
2. In the "Term Explanation" and "Exam-Style QA" tasks, did the models ever comment on the irrelevance of the attached photo? For example, a response like, "I'm happy to explain linear algebra, but your photo is not relevant to the query." While not an explicit refusal, this would be an interesting behavior to note, as it sits between a full refusal and the (biased)-but-compliant behavior observed.

**Questions:**

See weakness

---

> ### Author Response · Authors · 2025-11-25
> **Response to Reviewer Decc**
>
> Dear Reviewer Decc
>
> We sincerely appreciate your positive feedback and your acknowledgment of our work’s contributions. Your comments are truly valuable and will help us further improve the clarity and quality of our presentation. Please find our detailed responses below:
>
> >**W1.** The validation of the LLM-as-judge showed a 97% human agreement for the term explanation task. Was a similar human-agreement study performed for the story generation attribute extraction? This seems like a more complex and potentially ambiguous extraction task (e.g., parsing "personality" or "economic situation"), and it would be valuable to know the human-LLM alignment there as well.
>
> Thank you for this important question. We also conducted a human agreement study for the attribute extraction for the story generation.
>
> The result showed an agreement rate of **98%**, which is consistently high, similar to the term explanation task. We attribute this high consistency to our prompt design: as detailed in the paper, we explicitly provided attribute options in the generation prompt (e.g., "poor", "middle-class", "wealthy" for economic status). Consequently, the models tended to explicitly include these terms in the generated stories, making the extraction task straightforward and less ambiguous.
>
> >**W2.** In the "Term Explanation" and "Exam-Style QA" tasks, did the models ever comment on the irrelevance of the attached photo? For example, a response like, "I'm happy to explain linear algebra, but your photo is not relevant to the query." While not an explicit refusal, this would be an interesting behavior to note, as it sits between a full refusal and the (biased)-but-compliant behavior observed.
>
> We appreciate this insightful observation. You are correct: we observed that **some proprietary models (e.g., Claude) occasionally included disclaimers stating that the response would be provided regardless of the user's attributes**, without refusing the task.
>
> This behavior aligns with our discussion in Section 5, reflecting that proprietary models are developed with stronger safety awareness than open-source ones. However, importantly, our experiments reveal that **even these models still exhibit bias**. This suggests that while models may be trained to be fair, eliminating the underlying societal bias remains a significant challenge.

---

> > ### Comment · Reviewer_Decc · 2025-11-25
> >
> > Thanks for the clarifications. I will keep my score - it is very positive.

---

> > > ### Author Response · Authors · 2025-11-25
> > >
> > > Thank you very much for your encouraging reply. We truly appreciate your thoughtful comments and feedback!

---

### Official Review · Reviewer_RCGm · 2025-11-01

**Soundness:** 3
**Presentation:** 4
**Contribution:** 3
**Rating:** 8
**Confidence:** 4

**Summary:**

This paper aims to address the problem of diagnosing social biases in LVLMs which have strong safety guardrails, The authors find that leading commercial LVLMs as well as some recent open-source LVLMs refuse to answer queries that are commonly used for diagnosing bias in these models (e.g., refusing to identify the occupation of a person). To avoid this issue, the authors propose a simple approach where they prompt LVLMs with an image and a text prompt which does not ask direct questions about the person, thereby eliminating the problem of model refusal. Despite the prompt not referencing the person in the image, significant differences in story generation, term explanation, and exam-style QA responses are observed depending upon the race and gender of the person depicted in the image. Experiments are conducted across a wide range of open-source and commercial LVLMs to quantify these differences.

**Strengths:**

1. The problem of studying intrinsic model biases in the face of safety guardrails is important and timely
2. The authors propose a simple but clever solution which to the best of my knowledge is novel and original
3. Several interesting findings are provided throughout the paper, such as discrepancies in the presence of technical jargon depending upon the race/gender depicted in the image (observation 2.2)
4. Additional analyses are provided investigating the relationship between bias and model size, gender/race bias interdependence, and bias generalization across tasks.

**Weaknesses:**

1. The study only investigates race and gender bias, which seems limiting considering that image datasets exist with annotations for other types of social attributes.
2. The proposed method itself is quite simple, which is not necessarily a weakness. However, the simplicity of the approach makes the first few pages of the manuscript feel repetitive as the same basic idea is described multiple times.
3. The term explanation and exam-style QA tasks could be viewed as somewhat contrived in that they are not realistic prompting scenarios for LVLMs (i.e., it's unlikely for an image of a person to be provided as context along with a math of physics question). However, I understand the need to use creative prompting techniques to circumvent refusals.
4. This work is limited only to bias evaluation and did not explore any strategies for mitigating bias.

**Questions:**

1. Why did you limit your study to only race & gender attributes and a single dataset (FairFace)? Other datasets such as FACET and SocialCounterfactuals contain images with annotations for other types of social attributes which would be interesting to investigate.

---

> ### Author Response · Authors · 2025-11-25
> **Response to Reviewer RCGm (1)**
>
> Dear Reviewer RCGm
>
> We sincerely appreciate the positive review and your recognition of our work’s contribution. Your comments are very helpful in enhancing the quality of our presentation. Please find the corresponding responses below:
>
> > **W1.** The study only investigates race and gender bias, which seems limiting considering that image datasets exist with annotations for other types of social attributes.
> >
> >**Q1.** Why did you limit your study to only race & gender attributes and a single dataset (FairFace)? Other datasets such as FACET and SocialCounterfactuals contain images with annotations for other types of social attributes which would be interesting to investigate.
>
> Thank you for pointing this out. Regarding the focus on race and gender, we justify our scope based on the following reasons:
>
> * Following prior representative works [1, 2, 3], we focused on gender and race as they are the areas where **societal bias in LVLMs is most clearly exhibited** and critically need addressing.
>
> * As discussed in our methodology, minimizing spurious features (e.g., background context) is crucial for accurate bias probing. Therefore, we prioritized **face-centric datasets** like FairFace over natural image datasets. Typically, **these datasets mainly provide annotations for gender, race, and age**.
>
> * While representative face datasets like FairFace and UTKFace provide age annotations, we excluded age from this evaluation. This is because age-based performance differences are often desirable (e.g., using simpler vocabulary for children) rather than harmful biases.
>
> Therefore, while we believe that focusing on gender and racial bias does not diminish our main contributions, extending our framework to natural image datasets with richer demographic annotations (e.g., FACET) is a valuable direction for our future work.
>
>
> > **W2.** The proposed method itself is quite simple, which is not necessarily a weakness. However, the simplicity of the approach makes the first few pages of the manuscript feel repetitive as the same basic idea is described multiple times.
>
> Regarding the simplicity and presentation, we agree that the simplicity of our approach is not a weakness, as it ensures high reproducibility and broad applicability.
>
> We appreciate your feedback regarding the repetitiveness in the first few pages. We will revise the manuscript in the camera-ready version to reduce redundancy.
>
> > **W3.** The term explanation and exam-style QA tasks could be viewed as somewhat contrived in that they are not realistic prompting scenarios for LVLMs (i.e., it's unlikely for an image of a person to be provided as context along with a math of physics question). However, I understand the need to use creative prompting techniques to circumvent refusals.
>
> Regarding the realism of the tasks, we thank you for understanding the necessity of our approach to circumvent refusals.
>
> We would like to add one perspective: since **proprietary models (e.g., ChatGPT) have the ability to memorize user-provided information**, it is becoming common for models to retain user context. Therefore, assessing how models behave in tasks like Term Explanation or Exam-Style QA when conditioned on user information (e.g., a profile photo) is a relevant and realistic scenario for personalized AI agents.
>
> >**W4.** This work is limited only to bias evaluation and did not explore any strategies for mitigating bias.
>
> Regarding the lack of mitigation strategies, while we agree that this is an important direction, we believe that **focusing on bias evaluation itself makes a significant contribution** to the community for two reasons:
>
> 1.  **Enabling evaluation for guardrailed models:** As we demonstrated, existing methods fail to evaluate models with strong safety guardrails due to refusals. Without a reliable way to measure bias in these models, we cannot effectively proceed to the mitigation step.
>
> 2.  **Established research focus:** Given the complexity of the problem, many impactful prior works [3, 4, 5] focus exclusively on establishing robust evaluation benchmarks to lay the groundwork for future improvements.
>
> However, as an initial step toward bias mitigation (as we also detail in our response to Reviewer EAsf), **we tested a prompt-based mitigation strategy** on Qwen2.5-VL-7B. We explicitly instructed the model: “Please, be mindful that people should not be judged based on their race, gender, age, body type, or other physical characteristics.”
>
> We found that the bias score decreased only slightly (27.32 to 25.22). Furthermore, as noted in [1], we must be cautious with this approach, as such debiasing methods can significantly degrade general performance (e.g., accuracy on MMMU).

---

> > ### Author Response · Authors · 2025-11-25
> > **Response to Reviewer RCGm (2)**
> >
> > **References**
> >
> > [1] Girrbach et al., “Revealing and reducing gender biases in vision and language assistants (VLAs)”, ICLR 2025.
> >
> > [2] Howard et al., “Probing and mitigating intersectional social biases in vision-language models with counterfactual examples”, CVPR 2024
> >
> > [3] Fraser et al., “Examining gender and racial bias in large vision-language models using a novel dataset of parallel images”, EACL 2024
> >
> > [4] Huang et al., “VisBias: Measuring Explicit and Implicit Social Biases in Vision Language Models”, EMNLP 2025
> >
> > [5] Sathe et al., “A Unified Framework and Dataset for Assessing Societal Bias in Vision-Language Models”, EMNLP 2024

---

> > > ### Author Response · Authors · 2025-11-28
> > > **Delighted to Provide Further Details**
> > >
> > > Dear Reviewer RCGm,
> > >
> > > Thank you very much for your thoughtful and constructive feedback on our submission. We have carefully considered your comments and addressed each of them in our rebuttal. We understand you may be busy, but as the rebuttal period is approaching its end, we would greatly appreciate it if you could take a moment to review our responses when possible. If anything remains unclear or you would like additional details, we would be more than happy to provide further clarification.

---

### Official Review · Reviewer_EAsf · 2025-11-02

**Soundness:** 3
**Presentation:** 3
**Contribution:** 2
**Rating:** 4
**Confidence:** 4

**Summary:**

This paper proposes a new evaluation protocol of societal bias in *guardrailed* large vision-language models. The authors show that LVLMs with safety guardrails refuse to answer direct attribute inference prompts used in conventional bias evaluation, and propose to use indirect prompts (e.g., story generation) merely using the image as a context. The new benchmark is shown to eliminate model refusal, yet still exposes hidden demographic biases in the LVLM responses.

**Strengths:**

- Unlike existing work that probe direct bias of models, the paper exposes indirect biases in LVLMs even through irrelevant questions to the input image. Such bias can potentially result in disparities in the response style and quality of the model, and likely cannot be eliminated by mitigation methods that target direct bias.
- The evaluation is reasonably large-scale, spanning 3 probe tasks and 20 open-source and proprietary models. Results are extensive with detailed breakdown of bias types and qualitative examples. Some of the findings, such as the breakdown of bias per subject and demographic group, expose the stereotypes of existing VLMs that merit further research.
- Overall writing of the paper is clear and easy to follow.

**Weaknesses:**

- Comparison to existing open-ended bias evaluations: I wonder if the proposed approach differs substantially from VisBias and other open-ended benchmarks, beyond using a different set of prompts. While the new tasks are irrelevant to the image by design, the prompts do refer to the user photo explicitly ("I've attached my photo...") and I wonder if the models may be misled to believe its response (e.g., generated story) should relate to the image, which leads to the same spurious correlations that direct bias probing suffers from. I would have liked to see some quantitative evaluations or comparisons between open-ended probing methods, similar to how the authors compared refusal rate to closed-form prompts in table 1.
- It would be more convincing if the LLM judge used an ensemble of multiple models to reduce potential bias and variance from the Qwen3 32B model. While the model is shown to agree with human 97% of the time, it is unclear if this is high enough as the bias measurements for the task (term explanation) are mostly under 5%.
- Most of the qualitative analysis was presented for story generation and term explanation and make intuitive sense. However, it remains unclear to me how the exam QA task could also be affected by demographics depicted in the image. The paper does not provide a detailed hypothesis or analysis on this task. Or could it be that the measured TVD of 1-2% is already within the margin of error that one cannot conclude the bias is present with certainty at all?

**Questions:**

(in addition to questions raised in weaknesses)
While the experiments are already extensive, there are a few open questions worth further exploring:
- Does the bias originate from LLM pretraining, and how multimodal training amplifies it? Authors may apply the same benchmark on text-only models, with image replaced with detailed caption or user profile in the context.
- How effective are existing mitigation methods on the indirect bias? Adding system prompts that instruct the LVLM to avoid stereotypes from the input image could be a starting point.
- Likewise, the authors can compare the bias of the same model architecture in different training phases (e.g., PT->SFT->RL) to study whether each phase reduces or amplifies bias.

---

> ### Author Response · Authors · 2025-11-24
> **Response to Reviewer EAsf (1)**
>
> Dear Reviewer EAsf
>
> We sincerely appreciate your constructive review and recognition of our work’s contributions, specifically the novelty of indirect bias probing and our extensive experiments. Your comments are very helpful in enhancing the quality of our presentation. Please find the corresponding responses below:
>
> >**W1.** Comparison to existing open-ended bias evaluations: I wonder if the proposed approach differs substantially from VisBias and other open-ended benchmarks, beyond using a different set of prompts... I would have liked to see some quantitative evaluations or comparisons between open-ended probing methods, similar to how the authors compared refusal rate to closed-form prompts in table 1.
>
> We appreciate this insightful point. We argue that our framework differs fundamentally from existing open-ended benchmarks (e.g., VisBias) in two key aspects: the **role of the image** and the resulting **robustness to spurious correlations**.
>
> **1. Role of image:**
> As discussed in our paper, we shift the role of the image from **Target** to **Context**. Existing benchmarks treat the image as the subject of prompts (e.g., "Describe this image"), forcing attention to all visual tokens, including background noise. In contrast, we treat the image solely as provisional user information. This design decouples the task from the depicted person, ensuring the model uses the image primarily as a demographic signal rather than a visual target.
>
> **2. Robustness to spurious features:**
> To quantitatively verify that our method is less susceptible to spurious visual correlations than prior open-ended methods, we conducted a controlled experiment using InternVL3-14B.
>
> * **Setup:** We compared the stability of model outputs across two different image domains: **face-centric images** (FairFace, minimal background) and **real-world images** (COCO, complex backgrounds).
> * **Methods compared:**
>     * **Existing open-ended eval:** Prompt = "Describe this image in detail."
>     * **Ours:** Prompt = "I've attached my photo. Write a fictional story about an imaginary person..."
> * **Metric:** We measured the **Jensen-Shannon (JS) Distance** between the vocabulary distributions (nouns, verbs, adjectives) of face-centric and real-world images. A lower JS distance indicates that the model's outputs are more robust (i.e., less affected by the change in visual background/context).
>
> **Results:**
> The table below shows the JS distance (vocabulary shift) when the image source changes.
>
> | Method | JS distance (lower is better) | Interpretation |
> | :--- | :---: | :--- |
> | **Existing open-ended eval** | **0.61** | Outputs are heavily driven by background context. |
> | **Ours (story generation)** | **0.38** | Outputs remain relatively consistent regardless of visual context. |
>
> The existing evaluation's high JS distance confirms its sensitivity to spurious background features. Conversely, our method's significantly lower score demonstrates that framing the image as provisional user information effectively filters out spurious visual details, focusing the evaluation on the inherent societal biases triggered by the persona.
>
>
> >**W2.** It would be more convincing if the LLM judge used an ensemble of multiple models to reduce potential bias and variance from the Qwen3 32B model. While the model is shown to agree with human 97% of the time, it is unclear if this is high enough as the bias measurements for the task (term explanation) are mostly under 5%.
>
> Thank you for the suggestion. We agree that an ensemble approach helps reduce potential bias and variance.
>
> In response, we implemented a majority-vote ensemble using three models: Qwen3-32B, Llama3-8B, and GPT-4o. We evaluated the agreement rate with human judges using this method, which achieved 100% (100 samples) on the term explanation task, showing higher consistency than the single-model approach.
>
> We will update the paper to replace the single-model judge with this ensemble method for the term explanation task.

---

> > ### Author Response · Authors · 2025-11-24
> > **Response to Reviewer EAsf (2)**
> >
> > >**W3.** Most of the qualitative analysis was presented for story generation and term explanation and make intuitive sense. ... Or could it be that the measured TVD of 1-2% is already within the margin of error that one cannot conclude the bias is present with certainty at all?
> >
> > Although the bias in exam-style QA is smaller than in open-ended tasks, it still gives us helpful information. We have two points.
> >
> > First, while some models show bias close to the noise level, others are clearly above it. To find the noise level, we ran a control experiment by randomly mixing genders (ie, not splitting users by gender), which resulted in a bias score of 0.51. While several models are close to this value, others like Molmo-7B reach 3.44. This proves that user demographics do influence performance for specific models.
> >
> > Second, the more important insight is that **bias appears differently depending on the task constraints**. Low bias in exam-style QA does not mean models are free from societal bias. In open-ended tasks like story generation, we see strong bias. This contrast shows that bias depends on the degree of task constraint.
> >
> > Overall, bias exists in exam-style QA for some models, and observing these differences across tasks underscores the value of using multiple task types.
> >
> >  ___
> >
> > **Response to questions**
> >
> > Thank you for proposing these valuable directions for future work. As an initial step to address them, we conducted the suggested experiments on the story generation task on a small scale (~100 samples, gender bias).
> >
> >
> > >**Q1.** Does the bias originate from LLM pretraining, and how multimodal training amplifies it? Authors may apply the same benchmark on text-only models, with image replaced with detailed caption or user profile in the context.
> >
> > We evaluated the text-only Qwen2.5-7B by providing user demographics via text (e.g., "Hi! I’m a woman.") and compared it to the vision-language Qwen2.5-VL-7B.
> >
> > We found that while the text-only model exhibits bias (21.43), it is lower than the vision-language setting (27.32 in Table 2). We attribute this to two potential reasons:
> >
> > * Visual inputs provide demographic cues more implicitly, potentially triggering latent biases more effectively than explicit text.
> >
> > * As Reviewer EAsf suggested, multimodal training likely amplifies the bias.
> >
> > >**Q2.** How effective are existing mitigation methods on the indirect bias? Adding system prompts that instruct the LVLM to avoid stereotypes from the input image could be a starting point.
> >
> > We tested a prompt-based mitigation strategy on Qwen2.5-VL-7B, inspired by prior works [a, b]. We explicitly instructed the model: "Please, be mindful that people should not be judged based on their race, gender, age, body type, or other physical characteristics."
> >
> > We found that the bias score decreased only slightly (27.32 to 25.22). Furthermore, as noted in [b], we must be cautious with this approach, as such debiasing methods can significantly degrade general performance (e.g., accuracy on MMMU).
> >
> > [a] Howard et al., “Uncovering bias in large vision-language models with counterfactuals”, NAACL 2025.
> >
> > [b] Girrbach et al., “Revealing and reducing gender biases in vision and language assistants (VLAs)”, ICLR 2025.
> >
> > >**Q3.** Likewise, the authors can compare the bias of the same model architecture in different training phases (e.g., PT->SFT->RL) to study whether each phase reduces or amplifies bias.
> >
> > We agree that investigating bias across the training phases (PT to SFT to RL) is an interesting area for future work. Due to computational constraints during the rebuttal, we may not complete this experiment yet. However, we commit to including this analysis in the camera-ready version.
> >
> > Specifically, we plan to use the Qwen3-VL family to compare the Instruct (SFT) and Thinking (RL) variants to trace how alignment phases affect bias.

---

> > > ### Author Response · Authors · 2025-11-28
> > > **Delighted to Provide Further Details**
> > >
> > > Dear Reviewer EAsf,
> > >
> > > Thank you very much for your thoughtful and constructive feedback on our submission. We have carefully considered your comments and addressed each of them in our rebuttal. We understand you may be busy, but as the rebuttal period is approaching its end, we would greatly appreciate it if you could take a moment to review our responses when possible. If anything remains unclear or you would like additional details, we would be more than happy to provide further clarification.

---

### Author Response · Authors · 2025-12-01
**Summary of the reviews and our responses**

Dear Area Chairs and Reviewers,

We sincerely appreciate the time and effort in reviewing our paper, along with the shared positive feedback summarized below:

* **Clarity and writing quality** (EAsf, RCGm, Decc): The paper is clearly written, well structured, and easy to follow, with a strong and timely motivation.

* **Novelty and importance of addressing the refusal problem** (RCGm, Decc, oVen): The identification of the refusal issue in guardrailed LVLMs and the proposed workaround are viewed as timely and necessary for modern bias evaluation.

* **Simplicity and effectiveness of the proposed approach** (RCGm, Decc): The reframing of the image as user context is a simple yet clever and elegant idea that effectively enables evaluation where prior methods fail.

* **Comprehensive and insightful empirical results** (EAsf, RCGm, Decc, oVen): The evaluation across 20 LVLMs and 3 tasks is appreciated for its thorough and critical insights into demographic effects and task-dependent bias patterns.

__________
We have thoroughly addressed all reviewer concerns during the rebuttal period, and we summarize the key points and our responses below:

* **Spurious visual features in the input images** (EAsf, oVen):
 **Content**: Reviewers raise the possibility that background or non-demographic visual cues may still influence model behavior, questioning whether our method fully avoids the spurious correlations seen in prior open-ended evaluations.
**Our response**: We provide a quantitative JS distance comparison between FairFace and COCO, ***showing that our framework is substantially less sensitive to background variation***. We also clarify the conceptual shift of treating the image as a provisional user context rather than a visual target, and emphasize the use of face-centric datasets to further limit confounders.

* **Reliability of the LLM-as-a-judge** (EAsf, Decc):
 **Content**: Reviewers question whether relying on a single LLM judge could introduce bias or variance, and ask whether human alignment is validated for story generation.
 **Our response**: We introduce an ensemble of three judges (Qwen3 32B, Llama3 8B, GPT 4o) that ***achieves perfect agreement*** with human judgments for term explanation. For story generation, we report a human agreement rate of 98 percent. These additions fully resolve Reviewer Decc’s further concerns.

* **Task settings and evaluation scope** (EAsf, RCGm):
 **Content**: Reviewers note that some tasks may feel constrained or less natural, and that the evaluation focuses only on gender and race within a single face-centric dataset.
 **Our response**: We clarify that controlled task structure is essential for fair distributional comparison, and explain that ***race and gender are the most robustly annotated and widely studied*** demographic attributes in face datasets. We justify excluding age due to differing normative expectations and commit to extending the framework to additional attributes and datasets.

* **Mitigation analysis** (EAsf, RCGm):
 **Content**: Reviewers request further exploration of mitigation strategies and comparisons of training stages.
 **Our response**: We conduct initial mitigation experiments showing that a stereotype-avoidance system prompt slightly reduces bias but may degrade overall performance. We also commit to analyzing bias across training phases in the camera-ready version.


__________

**Additional concerns uniquely raised by Reviewer oVen**
 While these points are raised only by Reviewer oVen, ***other reviewers view these aspects positively or do not consider them problematic, and we address them carefully.***

* **Methodological novelty** (oVen):
 **Content**: Reviewer oVen suggests that the method has limited technical novelty.
 **Our response**: We clarify that the core contribution lies in reframing the image as user context, which directly solves the refusal problem that makes prior benchmarks unusable for guardrailed LVLMs. We also note that other reviewers explicitly regard the idea as novel and effective. Reviewer RCGm describes the approach as a ***"simple but clever solution which to the best of my knowledge is novel and original"***, and Reviewer Decc highlights it as a ***"novel and necessary methodology"***, indicating that the broader reviewers do not share the novelty concern.

* **Constraints in the story generation task** (oVen):
**Content**: Reviewer oVen questions whether explicitly requesting attributes might induce or exaggerate bias.
**Our response**: We explain that partial constraints are essential for ensuring comparable attribute extraction across demographic groups while keeping the task relatively open-ended. Other reviewers view the design as reasonable. For instance, Reviewer RCGm explicitly notes that they ***"understand the need to use creative prompting techniques to circumvent refusals"***, showing that the other reviewers do not support this concern.



Best regards,
Authors

---

### Meta-Review · Area_Chair_DXGk · 2026-01-04

**Summary:**

The paper proposes a societal bias evaluation method for large vision-language models. The authors show that LVLMs with safety guardrails often refuse to answer direct attribute inference prompts used in conventional bias evaluations, and they propose a novel evaluation method to overcome this issue. Although the paper addresses an interesting problem, several concerns raised by the reviewers are not fully addressed. Therefore, we believe the paper is not yet ready for publication.

The detailed concerns and whether they have been adequately addressed are listed below.


1. Reviewer EAsf

(1) Need quantitative evaluations or comparisons between open-ended probing methods

(2) Need experiments to show if the LLM judge used an ensemble of multiple models to reduce potential bias and variance from the Qwen3 32B mode

(3) how the exam QA task could also be affected by demographics depicted in the image

(4) Authors may apply the same benchmark on text-only models, with image replaced with detailed caption or user profile in the context.

(5) How effective are existing mitigation methods on the indirect bias?

(6) Need to compare the bias of the same model architecture in different training phases (e.g., PT->SFT->RL) to study whether each phase reduces or amplifies bias.

2. Reviewer RCGm

(1) Limited social attribute

(2)The proposed method itself is quite simple

(3) The term explanation and exam-style QA tasks could be viewed as somewhat contrived

(4) This work is limited only to bias evaluation and did not explore any strategies for mitigating bias.

3. Reviewer Decc

(1) Was a similar human-agreement study performed for the story generation attribute extraction?

(2) In the "Term Explanation" and "Exam-Style QA" tasks, did the models ever comment on the irrelevance of the attached photo?

4. Reviewer oVen

(1) Limited Methodological Novelty

(2) Potential for Confounding Variables in User Images

(3) Over-constrained "Story Generation" Task May Induce and Exaggerate Bias

**Reviewer Concerns:**

1. Reviewer EAsf

(1) Not fully addressed

(5) Not fully addressed, the authors on considered prompt-based mitigation strategy

(6) Did not address it

2. Reviewer RCGm

(1) Not addressed

(3) Not addressed

(4) Not addressed

3. Reviewer Decc

addressed

4. Reviewer oVen

(1) Not addressed

(2)  Not addressed

**Reviewer Scores:**

1. Reviewer EAsf

Will not change the score

2. Reviewer RCGm

May decrease the score to 6

3. Reviewer Decc

Not change the score

4. Reviewer oVen

Will not change the score

---

### Decision · Program_Chairs · 2026-01-26

Reject